# Sedentary Behavior and Lack of Physical Activity among Children in Indonesia

**DOI:** 10.3390/children10081283

**Published:** 2023-07-26

**Authors:** Laily Hanifah, Nanang Nasrulloh, Dian Luthfiana Sufyan

**Affiliations:** Faculty of Health Science, Universitas Pembangunan Nasional Veteran Jakarta, Jl. Raya Limo, Depok 16515, Indonesia

**Keywords:** children, physical inactivity, sedentary

## Abstract

Sedentary behavior and lack of physical activity among children in Indonesia is an important issue that needs to be addressed. It is estimated that 57% of children in Indonesia have insufficient physical activity. Studies have shown that children who engage in sedentary behaviors are at an increased risk for various negative health outcomes, including obesity, type 2 diabetes, cardiovascular disease, and poor mental health, compared to physically active ones. This article aims to provide recommendations to increase physical activity and reduce passive behavior in children in Indonesia. This is a commentary article developed from observing the recent progress of sedentary behavior and lack of physical activity among children in Indonesia and the potential consequences. The level of inactive behavior in children in Indonesia is relatively high. Factors that contribute to sedentary behavior and lack of physical activity among children in Indonesia are the increasing use of electronic devices and screen time, the lack of safe and accessible places to be physically active, the COVID-19 pandemic, as well as cultural and social norms that prioritize academic achievement over physical activity. To address sedentary lifestyles among children, there is a need for a comprehensive approach that addresses both the individual and societal factors contributing to the problem. This might include increasing access to healthy food options, promoting physical activity, and implementing education programs to raise awareness about the importance of healthy eating and physical activity, as well as limiting screen time.

## 1. Introduction

Childhood is crucial for developing physical, cognitive, and social skills. However, sedentary behavior and lack of physical activity among children in Indonesia is an important issue that needs to be addressed. Children’s lifestyles in Indonesia are becoming increasingly sedentary, with a growing trend of spending long hours sitting in front of screens and engaging in minimal physical activity [1,2]. As defined by the World Health Organization (2020), sedentary behavior is defined as any waking behavior characterized by an energy expenditure of 1.5 METS (metabolic equivalents; multiples of basal metabolic rates) or lower while sitting, reclining, or lying, while physical activity is defined as any bodily movement produced by skeletal muscles that requires energy expenditure [3]. The increase in energy expenditure and resulting challenge to pulmonary-cardiovascular systems have several beneficial effects on physical health and remarkably improved cardiovascular functioning.

It is estimated that 57% of children in Indonesia have insufficient physical activity [4]. The consequences of sedentary behavior and lack of physical activity among children in Indonesia are far-reaching and severe as this is a concern and could lead to long-term health problems for these children, impacting their academic performance and life satisfaction [5,6].

It is well known that regular physical activity is essential for children’s physical and mental health, and a lack of physical activity can lead to a range of adverse outcomes such as obesity, diabetes, and cardiovascular disease [7,8,9,10,11,12]. In addition to the health risks mentioned above, inactive children are likelier to experience poor academic performance, as physical activity improves cognitive function and concentration [13] Furthermore, lacking physical activity can lead to social and emotional problems, such as low self-esteem and depression [14].

One of the significant causes of sedentary behavior and lack of physical activity among children in Indonesia is the increasing reliance on technology and screens [8]. With the proliferation of smartphones, tablets, and other electronic devices, children spend more time sitting in front of screens than engaging in physical activities [15]. This trend is compounded by the fact that many schools in Indonesia need more facilities or programs for physical education, making it difficult for children to get the exercise they need [1]. The reasons to explore the scope of sedentary behavior and physical inactivity by using nationwide data before the pandemic suggested that there is a possibility that the trend of sedentary behavior and physical inactivity increasing during the pandemic leads to severe consequences. In this article, we will explore the extent of sedentary behavior and physical inactivity among children in Indonesia and the potential implications of this trend. We will also discuss possible strategies for promoting physical activity among children in Indonesia.

## 2. Methods

This is a commentary article developed from observing the recent progress of sedentary behavior and lack of physical activity among children in Indonesia and the potential consequences. This article systematically reviews the impact of sedentary lifestyles, such as obesity, diabetes mellitus, cardiovascular diseases, and mental health, then continues with the determinants of sedentary lifestyles (the increased use of electronic devices and screen time, change in dietary habits, lack of safe and accessible places for physical activity, cultural and social norms that prioritize academic achievement over physical activity and COVID-19 pandemic). Finally, this article discusses potential strategies for promoting physical activity among children in Indonesia.

## 3. Results and Discussion

Sedentary behavior contributes to physical inactivity, defined as insufficient physical activity to maintain good health. Sedentary behavior intensifies physical inactivity when one spends more time sitting than moving. Over time, physical inactivity may lead to severe life-threatening consequences [3].

Sedentary behavior among children will increase the risk of various health consequences, including obesity, cardiovascular alteration, bone density reduction and mental health problems [12,16,17,18,19]. An imbalance of energy intake and expenditure will lead to body fat accumulation, resulting in obesity. Obese children may develop cognitive problems, such as depression and anxiety, as they tend to become the object of bullying [20]. Sedentary behavior may also cause alteration in the cardiovascular system by increasing blood pressure, cholesterol level and risk of heart disease [21]. Children who maintain sedentary behavior are associated with a chance of type 2 diabetes and lower bone density later in adulthood [22,23].

Peltzer & Pengpid (2016) examined the relationship between physical inactivity frequency and sedentary behaviour among school children in the Association of Southeast Asian Nations (ASEAN) region towards 30,284 school children aged 13–15 years from seven ASEAN countries (Cambodia, Indonesia, Malaysia, Myanmar, Philippines, Thailand, and Vietnam) that participated in the Global School-based Student Health Survey (GSHS) between 2007 and 2013. The study found that the prevalence of physical inactivity was 80.4%, ranging from 74.8% in Myanmar to 90.7% in Cambodia and sedentary behaviour 33.0%, ranging from 10.5% in Cambodia and Myanmar to 42.7% in Malaysia. In multivariate analysis, factors such as not walking or biking to school, not attending physical education classes, inadequate vegetable consumption and lack of peer and parental or guardian support were associated with physical inactivity. It is also found that older age (14 and 15 years old), coming from an upper middle-income country, being overweight or obese, attending physical education classes, alcohol use, loneliness, peer support and lack of parental or guardian supervision were associated with sedentary behaviour [24].

The prevalence of insufficient physical activity for boys and girls in Indonesia slightly increased from 86.1% (2001) into 86.4% (2016) [25].This is in line with the global estimates that 81% boys and girls do not meet the WHO global recommendations on physical activity [26]. A scoping review conducted by Andriyani et al. (2020) of 166 studies worldwide found that the prevalence of sedentary behavior ≥ 3 h per day ranges between 24.5% to 33.8% [27].

Nationwide data related to Indonesian children’s physical activity available in the public domain is derived from Indonesia Basic Health Research 2007, 2013 and 2018. The relevant age group covered is 10 to 14 years. There is a slight decrease in the percentage of children’s lack of physical activity from 66.9% in 2007 to 64.4% in 2018. These numbers denote that more than half of Indonesian children were physically inactive, overlooking that regular physical activity helps maintain body weight and strengthen the cardiovascular system [28]. The 2013 data report the physical activity status provincially and not based on age group. However, only this report provides sedentary activity data. Among children aged 10 to 14, 28.2% did the passive activity for less than 3 h, 42.7% had sedentary activity for 3–5.9 h, and 29.1% were engaged in it for more than 6 h. Several regional studies reported the proportion of children engaged in sedentary behavior at Denpasar, Bali at 44.0% [29] and 27.1% in Gombara, Makassar [30]. A secondary data analysis derived from SEANUTS study (The South East Asian Nutrition Survey) reported 57.3% of Indonesian children aged 6–12 years engaged in TV/computer/Play Station watching more than 2 h a day [31].

It is well known that physical inactivity has a detrimental effect on health, accounting for 6% of global mortality [32]. Being physically inactive could lead to obesity. The accumulation of excess body fat characterizes obesity and can be conceptualized as the physical manifestation of chronic energy excess [33]. Obesity has become a public health problem in Indonesia, as seen by the increased prevalence of adult obesity from 10.5% in 2007 to 21.8% in 2018 [4]. Childhood obesity has become a growing concern in Indonesia over the past decade as the high level of inactive behavior and lack of physical activity among children in Indonesia. The proportion of obesity prevalence among children aged 5–12 years slightly increased from 8.8% in 2007 to 9.2% in 2018, whereas central obese prevalence among >18-years population increased drastically from 18.8% in 2007 to 31.0% in 2018.

The reasons for the increase in childhood obesity in developing countries, including Indonesia, are complex and multifaceted [11]. One major factor is the shift towards a more sedentary lifestyle, as children spend more time in front of screens and less time engaging in physical activity. Furthermore, there has been a change in dietary habits, with greater consumption of processed foods high in sugar and fat. Another contributing factor is the lack of access to healthy food options, particularly in rural and low-income areas. Many children in these areas rely on cheap, high-calorie foods, such as instant noodles and fried snacks, as their primary source of nutrition. Last but not least, the COVID-19 pandemic has led to increased sedentary behavior in children due to restrictions on physical activity.

Obesity is one of the most significant health risks associated with inactive behavior. A study published in the *American Journal of Epidemiology* showed that the prevalence of obesity increased with sedentary behavior, particularly among women [34]. The study suggested that reducing sedentary behavior and increasing physical activity could help to prevent obesity. A study conducted using the 2013 Indonesia Basic Health Research found that passive activity was correlated with overweight and obesity among those who lived in urban and rural areas [35].

Similarly, a lack of physical activity can lead to an increased risk of diabetes. According to a study published in Diabetes Care, sedentary behavior and physical inactivity are associated with a higher risk of type 2 diabetes [36]. The study recommended regular exercise to prevent and manage diabetes. Passive behavior is also linked to cardiovascular diseases. A European Journal of Epidemiology’s meta-analysis found that sedentary behavior was associated with an increased risk of cardiovascular diseases, including coronary heart disease, stroke, and heart failure [37]. The study suggested that reducing sedentary behavior and increasing physical activity can help to prevent cardiovascular diseases. In addition to physical health, passive behavior can also affect mental health. A review published in the Journal of Physical Activity and Health found that sedentary behavior was associated with a higher risk of depression, anxiety, and stress [18,38].

### 3.1. Determinants of Sedentary Lifestyles

There are several determinants of sedentary lifestyles, including the increased use of electronic devices, spending more time in front of screens and less time engaging in physical activity, change in dietary habits with greater consumption of processed foods, high-calorie foods, and foods high in sugar and fat, lack of safe and accessible places for physical activity, cultural and social norms that prioritize academic achievement over physical activity and COVID-19 Pandemic.

#### 3.1.1. The Increased Use of Electronic Devices and Screen Time

The increased use of electronic devices such as smartphones, computers, and televisions has been linked to sedentary behavior, which can adversely affect health. Sedentary behavior refers to sitting or lying down for extended periods while engaging in activities such as watching TV, using the computer or playing video games.

According to a study published in the Journal of the American Medical Association (JAMA), sedentary behavior has significantly increased among children and adults due to the increased use of electronic devices. The study found that from 2001 to 2016, the amount of time people spent sitting increased by an average of 1.5 to 2 h per day, with the majority of this increase being attributed to electronic devices [39].

Another study published in the journal Obesity Reviews found that sedentary behavior and the use of electronic devices were positively associated with obesity and overweight in children and adults. The study found that individuals who spent more time using electronic devices were likelier to have a higher body mass index (BMI) and increased risk the of developing obesity [40].

A study conducted by Tanjung et al. (2017) in Yogyakarta, Indonesia found that preschool children with high intensity use of gadgets are 1.3 times more likely to be obese [41]. This is inline with the findings from Uttari & Siddhiarta (2017) in Denpasar, Bali, Indonesia, revealing that there was a statistically significant relationship between the level of engagement in the screen time activities and the obesity in children with an odd ratio of 3.3 [42]. Another study conducted by Syahrul et al. (2016) in Indonesia also found that playing outdoors on weekends for less than 1 h were significantly associated with overweight children [43].

Furthermore, sedentary behavior and the use of electronic devices can also have negative impacts on mental health. A study published in the *Journal of Adolescence* found that excessive use of electronic devices, especially social media, was associated with poorer sleep quality and increased symptoms of depression and anxiety in adolescents [44].

The Council on Communications and Media of the American Academy of Pediatrics advises parents to limit children’s screen time to less than 2 h per day, to discourage screen media exposure, and to avoid placing televisions and internet-connected devices in children’s bedrooms [16]. Spending more time in front of screens and engaging less in physical activity are a growing problem in our modern society. With the rise of technology, people spend more time sitting in front of screens, whether for work, entertainment, or social media. Unfortunately, this sedentary lifestyle has serious consequences for physical and mental health [45,46].

Studies have shown that increased screen time is associated with decreased physical activity levels. It was found that adolescents who spent more time watching television or playing video games engaged in less physical activity than those who spent less time in front of screens [47,48,49]. This trend is not limited to adolescents, as adults who spend more time in front of screens also tend to have lower physical activity levels [12]. Other findings from 2527 children and adolescents (6–19 years old) from 2003/2004 and 2005/2006 National Health and Nutrition Examination Surveys (NHANES) found that high TV use was a predictor of high cardio-metabolic risk score (CRS) after the adjustment for MVPA and other confounders. Children and adolescents who watched TV ≥ 4 h per day were 2.53 times more likely to have high CRS than those who watched < 1 h per day.

Children who engage more in screen time activities may reduce physical activity and thus have a higher risk of obesity. A study in Kupang City, Indonesia found that screen-based activity for more than 2 h per day is particularly associated with an increased risk of obesity [50]. This is similar to the findings of research conducted in Yogyakarta, Indonesia where screen time of more than 2 h per day was associated with children being 2.6 more likely to be obese [51]. Higher screen time was also significantly associated with a higher level of energy intake.

There is also a relationship between the duration of gadget use and personal social skills in preschool-aged children. A review conducted by Oktafia et al. (2022) found that the gadget usage among pre school children increased from 38% in 2011 to 80% in 2015 and 13–18% of them experienced developmental issues [52]. Moreover, spending too much time in front of screens can adversely affect mental health. A study published in *JAMA Pediatrics* found that adolescents who spent more time in nets had a higher risk of developing symptoms of depression and anxiety. This suggests that screen time may contribute to mental health issues, particularly in vulnerable populations such as adolescents [53].

To mitigate the adverse effects of screen time, it is essential to prioritize physical activity and limit screen time. The World Health Organization recommends engaging in at least 150 min of moderate-intensity physical activity per week and limiting sedentary behavior [3]. This can be achieved through regular activity and minor changes, such as taking breaks from screens and engaging in more active pursuits, such as walking or cycling.

#### 3.1.2. A Change in Dietary Habits

Indeed, a significant relationship exists between sedentary behavior and a change in dietary habits towards greater consumption of processed foods, high-calorie foods, and foods high in sugar and fat. Research has shown that individuals who lead a sedentary lifestyle and consume a diet high in processed and high-calorie foods are at greater risk of obesity, type 2 diabetes, and cardiovascular disease.

A study published in the American Journal of Preventive Medicine found that sedentary behavior was positively associated with poor dietary habits in both men and women. The study found that sedentary behavior was linked to higher consumption of snacks, fast foods, sugar-sweetened beverages, and a lower intake of fruits and vegetables [54]. Individuals who spent more time sedentary were more likely to consume a diet high in processed foods and sugar-sweetened beverages [55]. Other study also found that those who spent more time inactive were less likely to consume a healthy diet, including fruits, vegetables, and whole grains [56,57].

A finding in 5 Southeast Asia countries (India, Indonesia, Myanmar, Sri Lanka and Thailand) showed that 76.3% of the 13 to 15 year-olds had insufficient fruits and vegetables consumptions (less than five servings per day); 28% reported consuming fruits and 13.8% consuming vegetables less than once per day, multivariate analysis found that sedentary behaviour and being overweight was protective of inadequate fruits and vegetable consumption [58].

Study conducted by Wulandari et al. in 2015 found that there was a correlation between energy intake and physical activity with overnutrition. Overweight and obesity can occur at any stage in life, including during primary school years. The prevalence of overnutrition in schoolchildren rose about 10.85% from 2007 until 2013, affecting both urban and rural areas. One of the key factors is the imbalance between energy intake and physical activity [59]. Therefore, promoting physical activity and healthy dietary habits should go synergistically to reduce the risk of chronic diseases associated with a sedentary lifestyle and unhealthy diet.

#### 3.1.3. Lack of Place for Physical Activity

Sedentary lifestyles and lack of safe and accessible places for physical activity are often interconnected. A sedentary lifestyle may result from limited access to safe and accessible areas for physical activity. In contrast, the lack of physical activity opportunities may contribute to sedentary behaviors. Studies have shown that individuals with limited access to safe and accessible places for physical activity are less likely to engage in physical activity [60].

The rapid economy improvement has reduced open spaces and facilities for physical activity or sports as well as parents permission for such activity. A qualitative study conducted by Roshita et al. (2021) in Indonesia, found that girls complained about obtaining permission from their parents to engage in outdoor activities due the parents being worried about their daughters’ safety and implementing stricter rules, to ensure that they return home before dark, thus limiting their physical activity after school [61].

A secondary analysis of Global School-Based Health Survey in Indonesia conducted by Yusuf et al. (2021) found that prevalence of active transportation by walking or bicycling to and from school among children was decreased from 47.2% (2007) to 32.3% (2015) [62]. Peer support among boys in 2015 was positively associated with low active transportation, meaning that the ownership by their peer of private vehicles would influence them too either to go together to and from school by using their own vehicle or by sharing a vehicle.

The study conducted by Has et al. with 130 pairs of school aged-children and their mothers/fathers, found that there was a significant correlation between children’s activity level, and access to safe housing and a playground with a sedentary lifestyle [63]. Therefore, improving access to safe and accessible places for physical activity may help to reduce health disparities and promote a healthier lifestyle. Initiatives such as community gardens, walking trails, and bike lanes can help to promote physical activity and improve access to safe and accessible places for physical activity [64].

#### 3.1.4. Social Norms

Cultural and social norms prioritizing academic achievement over physical activity may contribute to sedentary lifestyles. Studies have shown that individuals prioritizing academic success over physical activity are less likely to engage in physical activity [65,66].

The impact of sedentary lifestyles and cultural and social norms prioritizing academic achievement over physical activity on public health is significant. Cultural and social norms prioritizing academic achievement over physical activity may exacerbate the risk of non-communicable diseases by limiting opportunities for physical activity.

Furthermore, cultural and social norms prioritizing academic achievement over physical activity may disproportionately affect specific populations, including children and adolescents. Academic achievement is highly valued in many cultures, and physical activity may be seen as less important. This can lead to a lack of opportunities for physical activity and to a sedentary lifestyle among children and adolescents [65]. Yusuf et al. (2021) found the prevalence of physical activity among girls declined from 22.9 (2007) to 15.4% (2015) and boys from 26% (2007) to 17.6% (2015) in Indonesia. This is possibly related to time spent in school, from 7 h in 2007 to 8–10 h in 2015; thus, the time for physical activity was reduced due to more time spent for academic matters [62].

Improving access to physical activity is essential to promote a healthy and active lifestyle and reduce the negative impact of sedentary behaviors. Initiatives such as after-school sports, community sports leagues, and school-based physical education programs can promote physical activity and provide opportunities for children and adolescents to engage in physical activity [66].

#### 3.1.5. The COVID-19 Pandemic

The increasing number of COVID-19 cases requires countries affected to manage the transmission of COVID-19. The Government of Indonesia made a policy to accelerate the management of COVID-19. Implementing the policy of limiting community activities is enforced by enacting the Large-Scale Social Restrictions policy as an adaptation of the lockdown where all community activities are limited through a stay in their homes, except for essential or urgent activities. Until now, COVID-19 cases have declined compared to the first year of the pandemic. Although new to cases are still being found, there has yet to be an apparent shift from the pandemic to the endemic stage, and it requires countries to develop a resilient community [67].

The COVID-19 pandemic has led to increased sedentary behavior in children due to restrictions on physical activity, including school closures and limited opportunities for outdoor recreation [68]. A systematic review regarding physical activity in school children during the pandemic conducted by Ramadan (2022) found that 60–70% of school students did not meet the recommendation for physical activity [69]. A scoping review to explore the impact of COVID-19 on the movement behavior of children and youth conducted by Paterson et al. (2021) found that 150 studies consistently reported declines in physical activity, increased screen time and total sedentary behavior [70]. The pandemic is related to changes in the quantity and nature of physical activity and sedentary behavior among children and youth. A study in Yogyakarta, Indonesia, conducted by Andriyani et al. (2021), found that during the pandemic, mothers perceived their children to be less active and to use more screen-based devices, either for educational or recreational purposes, compared to before [71].

During the pandemic, children with higher levels of sedentary behavior had higher anxiety and depressive symptoms [72]. The findings were also supported by Pfefferbaum & Van Horn (2022), who stated, according to previous research, that decreased physical activity in the context of the COVID-19 pandemic and home confinement was associated with various psychological outcomes, including perceived stress, psychological distress, depression, anxiety, and hyperactivity inattention and prosocial behavior problems [18].

Promoting physical activity is essential to reduce the negative impact of sedentary behavior in children during the COVID-19 pandemic. Strategies to promote physical activity may include home-based physical activity programs, online physical education classes, and outdoor activities that adhere to social distancing guidelines [68]. In addition, encouraging parents to promote physical activity in their children may also be beneficial.

## 4. Conclusions and Recommendation

To address sedentary lifestyles among children, there is a need for a comprehensive approach that addresses both the individual and societal factors contributing to the problem. This might include increasing access to healthy food options, promoting physical activity, and implementing education programs to raise awareness about the importance of healthy eating and exercise. Policies to restrict the marketing and sale of unhealthy foods to children could also be implemented.

The high level of physically inactive behavior among children in Indonesia is a cause for concern, and interventions such as increasing access to physical activity facilities and programs, incorporating physical activity into the school curriculum, and addressing social and cultural barriers to physical activity can help to solve this issue. For example, providing safe and accessible places for children to be physically active, such as parks and playgrounds, community gardens, walking trails, and bike lanes, can encourage them to engage in regular physical activity. Schools should encourage regular physical education classes and promote active transportation to school, such as walking or cycling. Additionally, incorporating physical education classes and sports programs into the school curriculum can promote physical activity among children.

To mitigate the adverse effects of screen time, it is crucial to prioritize physical activity and limit screen time by taking breaks from screens and engaging in more active pursuits, such as walking or cycling. Parents could ask their children to do domestic tasks too to help their children away from the gadgets.

Furthermore, addressing social and cultural barriers to physical activity can also be important. For example, changing cultural and social norms prioritizing academic achievement over physical activity can help create a more supportive environment for children to be physically active. This can include promoting the importance of physical activity for overall health and well-being and encouraging a balance between academic success and physical activity. Furthermore, involving families and communities in promoting physical activity can create a supportive environment for children to be physically active. It is also recommended to conduct further studies to explore the impact of sedentary behavior and physical inactivity among Indonesian children to sharpen preventive action.

## Data Availability

The data are publicly available from various journals.

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
