# Peer review of "Sedentary Behavior and Lack of Physical Activity among Children in Indonesia"

_children, 2023, doi:10.3390/children10081283_

Round 1

Reviewer 1 Report (Previous Reviewer 4)

All comments I wrote have been addressed. The manuscript has improved, congratulations.

Author Response

Thank you

Reviewer 2 Report (New Reviewer)

Thank you for allowing me to review this commentary on sedentary behavior of children in Indonesia. The writing was easy to follow and understand with a few minor edits required for English. I do not think the Methods section belongs since it is a repeat of previous information, and it does not truly describe methods. Results does not seem like an appropriate header either, as there are no results reported. Lines 87-15 seem like a repeat from earlier and/or irrelevant because the information does not focus on Indonesia. 

The information presented in lines 222-401 is information that is already known. I see the purpose of this commentary being sharing of previous knowledge (what we already know) with the majority of the information focusing on the statistics from Indonesia specifically, and furthermore, what can or should be done about it. I was left with very little of this information. I encourage the authors to focus more on the current state of affairs and most of the manuscript on solutions.

This aspect of the manuscript was well done with a few grammatical changes needed.

Author Response

This is a commentary article, not original research thus we did not collect primary data and only used published articles derived from various type of data, some of those are nationwide data and the others are local data. 

Yes, the information presented in lines 222-401 (now changed into lines 230-409) may nothing new, but there are no such publications that gathered all data and we compile researches conducted worldwide with highlight in Indonesia.  The research regarding evidences of sedentary behaviours on children in Indonesia are still limited. Thus, this article tried to collect information from the international and local journals and sometimes the result obtained from a regional study. So that is the novelty of this article and should be published. We did focus on the current state as describe in line 163-195 and wrote solutions in conclusions and recommendation section.

Reviewer 3 Report (New Reviewer)

Dear authors,

Thank you very much for your work. However, I do have some questions. Please find my specific comments below: 

Comment 1.   

Abstract:The abstract should follow the style of structured abstracts, but without headings: 1) Background: Place the question addressed in a broad context and highlight the purpose of the study; 2) Methods: Describe briefly the main methods or treatments applied. Include any relevant preregistration numbers, and species and strains of any animals used; 3) Results: Summarize the article's main findings; and 4) Conclusion: Indicate the main conclusions or interpretations. The abstract should be an objective representation of the article: it must not contain results which are not presented and substantiated in the main text and should not exaggerate the main conclusions.

It is recommended to add the study methodology to the abstract. 

Comment 2.     

 In the introduction, the manuscript provides an introduction to the significance of the selected topic, that is, the important effects of sedentary behavior and lack of physical activity on children. The study noted that the aim of the study was to explore the extent of sedentary behavior and physical inactivity among Indonesian children and the potential consequences of this trend. However, I did not find a review of existing literature on this area of research. For example, has the literature discussed the current status and impact of sedentary behavior and physical inactivity among Indonesian children. If so, the revised manuscript suggests citing the literature and describing the innovations of your study. If not, it is also recommended to add a description.

Comment 3.     

Results and Discussion :In the analysis of the determinants of sedentary lifestyles, it was suggested that these two factors be discussed together, namely, increased time spent using electronic devices and increased time spent in front of screens. This is because these two factors have commonalities.

Comment 4.     

Results and Discussion :Please explain what this paragraph is trying to convey.

World Health Organization data shows that from an estimated 36 million deaths, or 63% of the 57 million deaths that occurred globally were due to noncommunicable diseases, comprising mainly cardiovascular diseases (48% of noncommunicable diseases), cancers (21%), chronic respiratory diseases (12%) and diabetes (3.5%) where 80% of all deaths (29 million) from non-communicable diseases occurred in low- and middle-income countries, and a higher proportion (48%) of the deaths in the latter countries are premature (under the age of 70) compared to high income countries (26%)………. (line 104-123).

Comment 5.     

Supplementary Materials: I did not find the supplementary material in the system.

Figure S1. Prevalence of insufficient physical activity among population aged 11-17 years globally and based on income group in 2001 and 2016,

Figure S2. Prevalence of insufficient physical activity among boys aged 11-17 years in 2016,

Figure S3. Prevalence of insufficient physical activity among girls aged 11-17 years in 2016

 Minor editing of English language required. 

Author Response

Comment 1.   

Abstract:The abstract should follow the style of structured abstracts, but without headings: 1) Background: Place the question addressed in a broad context and highlight the purpose of the study; 2) Methods: Describe briefly the main methods or treatments applied. Include any relevant preregistration numbers, and species and strains of any animals used; 3) Results: Summarize the article's main findings; and 4) Conclusion: Indicate the main conclusions or interpretations. The abstract should be an objective representation of the article: it must not contain results which are not presented and substantiated in the main text and should not exaggerate the main conclusions.

Respond (R): Yes, we insert the headings, such as background and add study methodology to the abstract

Comment 2.     

 In the introduction, the manuscript provides an introduction to the significance of the selected topic, that is, the important effects of sedentary behavior and lack of physical activity on children. The study noted that the aim of the study was to explore the extent of sedentary behavior and physical inactivity among Indonesian children and the potential consequences of this trend. However, I did not find a review of existing literature on this area of research. For example, has the literature discussed the current status and impact of sedentary behavior and physical inactivity among Indonesian children. If so, the revised manuscript suggests citing the literature and describing the innovations of your study. If not, it is also recommended to add a description.

(R): Yes, there are no existing studies or literatures discussed the impact of sedentary behavior and physical inactivity among Indonesian children. Nationwide data we had only presents the percentage of children's lack of physical activity, duration of sedentary activity and prevalence of children obesity. Therefore, we recommend scholars and researchers to do so.

Comment 3.     

Results and Discussion ï¼šIn the analysis of the determinants of sedentary lifestyles, it was suggested that these two factors be discussed together, namely, increased time spent using electronic devices and increased time spent in front of screens. This is because these two factors have commonalities.

(R): Yes, we changed and discussed it together.

Comment 4.     

Results and Discussion ï¼šPlease explain what this paragraph is trying to convey.

World Health Organization data shows that from an estimated 36 million deaths, or 63% of the 57 million deaths that occurred globally were due to noncommunicable diseases, comprising mainly cardiovascular diseases (48% of noncommunicable diseases), cancers (21%), chronic respiratory diseases (12%) and diabetes (3.5%) where 80% of all deaths (29 million) from non-communicable diseases occurred in low- and middle-income countries, and a higher proportion (48%) of the deaths in the latter countries are premature (under the age of 70) compared to high income countries (26%)………. ï¼ˆline 104-123).

(R): The paragraph trying to convey the magnitude of the non-communicable diseases (NCD) globally where the deaths due to NCD in LMICS happened under 70 years of age as stated in the previous sentences (line 107-111) below:

Sedentary behavior may also cause alteration in the cardiovascular system by increasing blood pressure, cholesterol level and risk to heart disease21. Children who maintain sedentary behavior are associated with risk of type 2 diabetes and lower bone density later in adulthood22,23

Comment 5.     

Supplementary Materials: I did not find the supplementary material in the system.

(R):The name of the file is List of figures.pdf consisted of all figures in the main document.

Round 2

Reviewer 3 Report (New Reviewer)

The revised manuscript is much better.

Author Response

Response to reviewer comments:

Reviewer 3 asked to remove the titles in the abstract

Response 1: Yes, we removed the titles in the abstract

Reviewer 2 asked to more focus on Indonesia data and minimise global data within the manuscript.

Response 2: Yes, we deleted lines 145-149

We added lines 294-298

Reviewer 2 asked to focus on solutions for Indonesian youth, not only in the conclusion but also in discussion section

Response 3: We added lines 439-440 in the conclusion and recommendation and actually we did write solutions at the end of the determinants, for example

  1. The increased use of electronic devices and screen time

We wrote the solutions in line 300-306

To mitigate the negative effects of screen time, it is important to prioritize physical activity and limit screen time. The World Health Organization recommends to engage in at least 150 minutes of moderate-intensity physical activity per week and limit sedentary behavior49. This can be achieved through regular activity as well as small changes such as taking breaks from screens and engaging in more active pursuits, such as walking or cycling.

  1. A change in dietary habits

We wrote the solutions in line 324-236

Therefore, promoting physical activity and healthy dietary habits should go synergistically to reduce the risk of chronic diseases associated with a sedentary lifestyle and unhealthy diet.

  1. Lack of place for physical activity

We wrote the solutions in line 343-347

Improving access to safe and accessible places for physical activity may help to reduce health disparities and promote a more active and healthier lifestyle. Initiatives such as community gardens, walking trails, and bike lanes can help to promote physical activity and improve access to safe and accessible places for physical activity55.

  1. Social norms

We wrote the solutions in line 365-370

Improving access to physical activity is essential to promote a healthy and active lifestyle and reduce the negative impact of sedentary behaviors. Initiatives such as after-school sports programs, community sports leagues, and school-based physical education programs can help to promote physical activity and provide opportunities for children and adolescents to engage in physical activity57.

  1. The COVID-19 pandemics

We wrote the solutions in line 407-413

Promoting physical activity is essential to reduce the negative impact of sedentary behavior in children during the COVID-19 pandemic. Strategies to promote physical activity may include home-based physical activity programs, online physical education classes, and outdoor activities that adhere to social distancing guidelines59. In addition, efforts to encourage parents to promote physical activity in their children may also be beneficial.

This manuscript is a resubmission of an earlier submission. The following is a list of the peer review reports and author responses from that submission.

Round 1

Reviewer 1 Report

Overall, this is reasonably well written and involves a valuable and interesting topic and I appreciate the author's efforts in addressing it. However, the structure and varying terminology makes the flow of reading quite difficult. It switches between 'exercise', 'sedentary behaviour', 'physical inactivity' 'inactive behaviour' and 'obesity' which makes the story unclear.  I have recommended some amendments that I think will help with the clarity of the paper.

Title - lack of physical activity rather than exercise is more appropriate. Exercise and physical activity are used interchangeably throughout but they are different things (exercise is structured activity). I think this should consistently be 'physical activity' because this is what is referred to and defined in the introduction. Obesity could be mentioned in the title due to some specific focus on this in the article as well? Is it a review of sedentary lifestyle (ie sedentary behaviour and physical inactivity) and obesity?

Abstract - I think starting with a statement regarding obesity is confusing considering the title. Physical inactivity has risk factors irrespective of obesity. Line 11 - increase physical activity (rather than exercise). 

Line 13 - sedentary behaviour is different to physical inactivity. You could have sufficient physical activity levels but have too much sedentary time and still be at risk. I realise this is defined further along but it is not clear at this point. I also realise that sedentary lifestyle is defined on Line 319 well into the review so it probably makes sense to define that in the introduction as that seems to be the combination of sedentary behaviour and physical inactivity.

Line 19 - physical activity recommendations rather than exercise.

Introduction - overall this is clearly written and leads to the aim. Although physical activity is defined, sedentary behaviour should be as it's not just those not meeting the physical activity recommendations, sedentary time is different and a key aim of this study. I have just realised that it is defined along with physical inactivity in the discussion (lines 81-87) so should be moved up to the introduction to be clear on the terminology (which is also the same for sedentary lifestyle. I believe this article focusses on physical activity, physical inactivity and sedentary behaviour (combined as sedentary lifestyle), and obesity (not exercise).

Line 56 - although screen time is a key factor, it might be worth adding in some other identified factors that could impact on physical activity levels and sedentary time.

Methods - I think it is a little bit confusing with interchangeable use of sedentary lifestyle and physical inactivity. As I said previously, terminology should be reviewed and clarified.

Line 72- how did the article develop systematically? 

Results and discussion - There is some interesting and well thought out detail in this section. It is clear to read although but the structure could be amended slightly (mostly near the beginning) to create a better flow for the reader.

Line 81-88 I think it would make sense to move these definitions earlier and then possibly subhead the sections 'Sedentary Behaviour' and then 'Physical Inactivity' as it gets quite confusing with them combined. Or the heading of Sedentary Lifestyle would include both?

From line 89 - this is clear that is it sedentary behaviour and not physical inactivity. However, figures 1, 2 and 3 are then related to stats regarding physical inactivity so then is out of context. It is worth adding a linking paragraph to introduce physical inactivity as a new section.

Line 169 - 'physical inactivity could lead to obesity'. I think this section might as a separate heading of 'obesity'. There is quite a lot on this and wonder if if should be also in the title? As a definition, this might sit better in the introduction so all terms are defined at the start.

Line 215- should it be determinants of sedentary lifestyle and obesity?

Line 223- these subheadings should be the next level (ie are they under the determinants section?)

Line 319 - if sedentary lifestyle is defined to combine sedentary behaviour and low levels of physical activity then it might be useful to use this term more regularly from the start to avoid having to constantly state 'sedentary behaviour' and 'physical 'inactivity as different entities (unless the research specifically looks at one or the other.

Line 398- this sentence does not make sense.

Reviewer 2 Report

Given the importance of this work for its practical implications, I think it is appropriate to highlight some issues that do not facilitate its reading. Overall, the document is confusing and repetitive. For example, the damage from a sedentary life is reported in too many different parts and this reduces the flow of speech. It is advisable to define a limited section in which to talk about such negative consequences, avoiding to include them in all parts of the text. Other concepts are repeated, I will mention some of them, but it would be appropriate to revise the whole text.

- lines 10-12: Move the study objective to line 25.  After listing the issues related to physical inactivity, it is easier to understand the goal of the study

-lines 57-58: To cite studies or statistics that confirm this dependence. If not present, rewrite the text stating that this dependence is a hypothesis of the authors.

-line 88: It might be useful to mention here the WHO guidelines as an international reference.

-lines 163-166: already stated earlier.

-lines 180-190: Specify if these are authors' assumptions or if there are studies or data that confirm the existence of these factors.

-lines 261-264: Studies should be mentioned.

-lines 274-276: already stated earlier.

-lines 279-280: already stated earlier.

-line 292: Studies, as reported, seem to show that sedentary behaviour is a determining factor for the acquisition of negative eating habits. On line 218, however, it is claimed that poor nutrition is a determining factor of  a sedentary life. Therefore, only studies consistent with this claim should be reported. 

-lines 319-320: already stated earlier.

-lines 366-368: already stated earlier.

-lines 395-398: already stated earlier.

Reviewer 3 Report

Dear Authors,

The title of your manuscript is adequate and interesting and triggered my curiosity to get more knowledge of the status of physical activity and exercise among Indonesian children- see attached file

Reviewer 4 Report

The information on lines 23 to 34 must be supported by references.

References 1 and 2 must be before point. This occurs in all references.

IN the introduction the risks that sedentary behaviors and lack of physical activity bring are exposed. In the final part, from line 63, they expose the objective of their research, but the reason (the problem) for which it is necessary to explore the scope of sedentary behavior and the lack of possible consequences of this trend is not clear. Also discuss possible strategies to promote physical activity and exercise among Indonesian children.

They should add more information that supports the conduct of their investigation.

Results and discussion

The information contained in this section seems appropriate to me, since it exposes the consequences of sedentary behavior, physical inactivity, in addition to providing recommendations. However, the problem I see is that they have "nothing new", it is a compilation of information from several important sources.

Also, which of all the information they provide is particularly for Indonesian children? Because what they provide is what children all over the world should do, regardless of the country where they live.

In summary, I think that background information supporting the need to carry out this research should be included in the introduction.

Besides. they must incorporate the particularities of Indonesian children (what differentiates them from other children in the world), and based on these particularities deliver the information.